# A Comprehensive Metabolomic Analysis of Volatile and Non-Volatile Compounds in *Folium Artemisia argyi* Tea from Different Harvest Times

**DOI:** 10.3390/foods14050843

**Published:** 2025-02-28

**Authors:** Hui Wu, Liya Niu, Jiguang Chen, Haixia Xu, Cailin Kong, Jianhui Xiao

**Affiliations:** School of Food Science and Engineering, Jiangxi Agricultural University, 1101 Zhimin Road, Nanchang 330045, China; hui13479353118@163.com (H.W.); nly8483@163.com (L.N.); chenjiguang@jxau.edu.cn (J.C.); hxxu@jxau.edu.cn (H.X.)

**Keywords:** *Folium Artemisia argyi* tea, harvest time, flavor profiles, metabolomics analysis, antioxidant capacity

## Abstract

To develop and utilize *Folium Artemisia argyi* (FAA) tea resources, UPLC-MS/MS, HS-GC-IMS, and HS-SPME/GC×GC-TOFMS were adopted to analyze its volatile and non-volatile compounds, when harvested from March to June, in combination with its antioxidant activity. Here, 1742 volatile compounds and 8726 non-volatile compounds were identified, with 75 differential volatile metabolites and 36 key flavor compounds screened. Notably, 1-octen-3-one, (E)-2-octenal, (E)-2-undecenal, and heptanal were identified as major contributors to the sweet, fruity, green, and herbal aromas, and the concentration of them was highest in June-harvest FAA tea. Furthermore, metabolomics revealed that there were 154 non-volatile differential metabolites in FAA tea at four harvest times, which were mainly related to amino acid biosynthetic pathways. Samples harvested in June also showed the strongest antioxidant capacity, which was positively correlated with D-xylitol, L-glutamic acid, honokiol, and costunolide. These findings highlight June as the optimal harvest time, providing FAA tea with superior flavor and enhanced antioxidant properties, underscoring its potential as a valuable resource for functional food development.

## 1. Introduction

*Folium Artemisia argyi* (FAA), derived from the leaves of the perennial plant *Artemisia argyi*, has been widely used as an herbal medicine for centuries from a historical perspective. It is renowned for its diverse therapeutic effects, including antibacterial, antiviral, anti-asthmatic, antitussive, expectorant, and anticancer properties [1,2]. Simultaneously, rich in various active compounds, such as essential oils, organic acids, flavonoids, polysaccharides, terpenoids, amino acids, coumarins, and other foodborne phytochemicals [3], FAA has been scientifically proven to offer significant health benefits, including antioxidation [4], anti-fatigue [5], and obesity prevention [6]. Given the high nutritional value and unique aroma, FAA is widely applied as a food ingredient in the fields of tea and other food products. Specifically, the tea made from FAA (called FAA tea) is highly favored in Asian countries, like China, Japan, and South Korea, for its unique flavor.

Despite prior studies on the biochemical components and biological characteristics of FAA [1,7], research focusing on the flavor of FAA tea is rarely reported. Key gaps include the absence of standardized harvest times and a lack of understanding regarding the variation in flavor metabolites during its growth and development. Up to now, it is generally believed that harvest times of plants have significant effects on the bioactive components and quality of plant-derived compounds. Zeng et al. observed higher amino acid and catechin levels in early spring tea compared to other harvest periods [8], while Chen et al. found that the phenolic content of purslane in later harvests was up to 30 times higher than in earlier ones [9]. Similarly, Liu et al. reported seasonal variations in volatile compounds of *Citrus reticulata ‘chachi’* peel, with green flavors dominating from July to October and fruity notes from November to December [10]. These studies suggest that harvest timing could critically affect the flavor and metabolite profile of FAA tea.

Obviously, emerging metabolomics technologies have allowed for the simultaneous detection of hundreds of endogenous metabolites and have been widely used in plant physiology, stress tolerance, phytochemistry, and flavor studies [11,12], as well as in food science [13]. Zhang et al. utilized LC/MS-based metabolomics and GC/MS analysis to improve the taste and aroma of Keemun black tea through solid-state fermentation [14]. Liu et al. characterized the floral and chestnut aromas in green tea using GC-MS and HS-GC-IMS [15]. Sai et al. used UPLC-MS/MS to quantify hydrated catechins, caffeine, gallic acid, theanine, and theaflavins in tea plants [16]. These references demonstrate the potential of metabolomics in flavor studies and its applicability to FAA tea.

Based on this, ultra-performance liquid chromatography-tandem mass spectrometry (UPLC-MS/MS), headspace gas chromatography-ion mobility spectrometry (HS-GC-IMS), and headspace solid-phase microextraction two-dimensional gas chromatography time-of-flight mass spectrometry (HS-SPME/GC×GC-TOFMS) were employed to analyze the non-volatile and volatile compounds of FAA tea across four harvest periods (March, April, May, and June). Principal component analysis (PCA) and orthogonal partial least squares discriminant analysis (OPLS-DA) were used to compare component differences among samples, while antioxidant assays evaluated the in vitro antioxidant capacity of FAA tea. To our knowledge, this is an important comprehensive investigation into the flavor characteristics and metabolite profiles of FAA tea at different harvest times. The findings will provide valuable data and theoretical insights for optimizing the harvest period to achieve the best flavor in FAA tea.

## 2. Materials and Methods

### 2.1. Materials

The FAA samples analyzed in this study were systematically collected in March, April, May, and June, which were provided by Jiangxi Yuai Agriculture Co., Ltd. (Zhangshu, China). These Artemisia argyi grow in the Artemisia argyi growing base in Wutang Village, Linjiang Town, Zhangshu City, Jiangxi Province (115°20′–24′ E, 27°54′–04′ N), and are characterized by similar growth patterns and the absence of pests and diseases. The samples were identified by the Teaching and Research Section of Resources and Development, College of Agronomy, Jiangxi Agricultural University, as those of *Artemisia argyi Lévl. et Vant* of the Compositae family, and voucher specimens have been deposited in the Jiangxi Agricultural University for future reference. The samples were stored at −80 °C until further analysis to ensure their stability and prevent degradation. All chemical reagents were of analytical grade and purchased from Sigma Aldrich (Shanghai, China).

### 2.2. Appearance Measurement

The FAA tea samples, produced by drying FAA harvested in March, April, May, and June, were designated as FAA-3, FAA-4, FAA-5, and FAA-6, respectively. For preparation, 3 g of each FAA tea sample was steeped in 200 mL of boiling water for 10 min and then filtered to obtain the FAA tea infusion. The color of FAA tea soup was evaluated using a Hunter Color Quest XE (Hunter Lab, Reston, VA, USA) based on L* (+ white, -−black), a* (+ red, −green), and b* (+ yellow, −blue). This method was adapted from Johnson et al. with some modifications [17].

### 2.3. Antioxidant Capacity Measurements

#### 2.3.1. DPPH• Radical Scavenging Assay

The experimental method was based on the protocol described by Araceli et al. with some modifications [18]. The FAA tea powder was prepared by freeze-drying fresh tea leaves, followed by grinding and sieving through a 100-mesh sieve to obtain a fine powder. Specifically, 0.1 g of FAA tea powder was added to 5 mL of anhydrous ethanol. Although the tea powder did not completely dissolve, the mixture was vigorously vortexed to ensure homogeneity, followed by centrifugation at 10,000 r/min for 15 min [19]. Then, 0.5 mL of supernatant was mixed with 2 mL of a 0.2 × 10^−5^ mol/L DPPH• free radical solution and incubated in the dark for 30 min at 25 °C. The absorbance of the resulting solution was measured at 515 nm using a UV-VIS spectrophotometer. DPPH• free-radical scavenging activity (%) is measured based on the following equation.Scavenging activity (%) = [1 − (A_sample/_A_0_)] × 100(1)
where A_sample_ and A_0_ are the absorbance measured at 515 nm for the sample and control, respectively. IC_50_ refers to the concentration of effective phenols at 50% removal of DPPH• free radicals.

#### 2.3.2. ABTS•^+^ Radical-Scavenging Assay

ABTS•^+^ radical-scavenging activity was determined using the method described by Li et al. with minor modification [20]. The sample preparation for the ABTS•^+^ assay was the same as described in Section 2.3.1 for the DPPH• assay. The free radical generation of ABTS•^+^ was completed by blending 2 mL of a 7.4 mmol/L ABTS•^+^ solution with 7.15 mL of a 2.6 mmol/L potassium persulfate solution and avoiding light at 25 °C for 12 h; 100 μL of the sample supernatant was mixed with 1 mL of the ABTS•^+^ solution and located at 25 °C in the dark for 30 min. The blank was replaced with 100 μL of anhydrous ethanol.Scavenging activity (%) = [1 − (A_sample/_A_0_)] × 100(2)
where A_sample_ and A_0_ are the absorbance measurements of the sample and the control at 734 nm, respectively. IC_50_ refers to the concentration of effective phenols at which ABTS•^+^ free radicals are removed by 50%.

#### 2.3.3. Ferric-Reducing Antioxidant Power (FRAP) Assay

FRAP was performed with the method described by Zhao et al. with slight modifications [21]. The sample preparation for the FRAP assay was the same as described in Section 2.3.1 for the DPPH• assay. FRAP reagent was prepared by blending sodium acetate buffer (300 mmol/L, pH 3.6), TPTZ solution (10 mmol/L, 40 mmol/L HCl as solvent), and FeCl_3_·6H_2_O solution (20 mmol/L) at a 10:1:1 volume proportion. The FRAP reagent (3 mL) was mixed with the above supernatant solution (1 mL), and an equal amount of ethanol was taken as the reagent blank. The absorbance of the mixture at 593 nm was determined after the reaction in a place without light for 30 min. A calibration curve was established using the Fe (II) solution (y = 0.9893x − 0.0999, R^2^ = 0.9995) with known concentrations of 0.15 to 1.5 mmol/L (FeSO_4_·7H_2_O), and the results were expressed as mmol/L Fe^2+^/g dry weight (DW).

### 2.4. UPLC-MS/MS Detection of the Nonvolatile Compounds

#### 2.4.1. Sample Preparation

FAA tea samples were freeze-dried using a vacuum freeze-dryer (H1850-R, Hunan Xiangyi Laboratory Instrument Development Co., Ltd., Changsha, China) and accurately weighed to 50 mg ± 1.7 mg. The weighed samples were placed in a 2 mL centrifuge tube, and 600 µL of methanol containing 2-chloro-L-phenylalanine (4 ppm) was added using a calibrated micropipette. The mixture was vortexed for 30 s, ground at 55 Hz for 60 s, and then centrifuged at 12,000 rpm for 10 min at 4 °C. The obtained supernatant was filtered through a 0.22 μm membrane and subsequently transferred to a 2 mL sample vial for LC-MS analysis [22].

#### 2.4.2. UPLC-MS/MS Detection

The chromatography was performed on ACQUITY UPLC^®^ HSS T3 (2.1 × 100 mm, 1.8 µm) (Waters, Milford, MA, USA) column with a flow rate of 0.3 mL/min at a column temperature of 40 °C. The sample size was 2 μL. Each sample was analyzed separately in positive ion mode and negative ion mode to ensure optimal data quality. In the positive ion mode, the mobile phase consisted of 0.1% acetonitrile formate (B_2_) and 0.1% formic acid water (A_2_), and the gradient elution procedure was 0–1 min 8% B_2_, 1–8 min 8–98% B_2_, 8–10 min 98% B_2_, 10–10.1 min 98–8% B_2_, and 10.1–12 min 8% B_2_. In the negative ion mode, the mobile phase consisted of acetonitrile (B_3_) and 5 mM ammonium formate water (A_3_), and the gradient elution procedure was 8% B_3_ for 0–1 min, 1–8 min 8–98% B_3_, 8–10 min 98% B_3_, 10–10.1 min 98–8% B_3_, and 10.1–12 min 8% B_3_ [23]. An Orbitrap Exploris 120 mass spectrometry detector (Thermo Fisher Scientific, Waltham, MA, USA) and electrospray ion source ESI positive and negative ion mode were used to collect data, respectively. The positive (+) and negative (−) ESI modes parameters were as follows: ion spray voltage 3.50 kV (+) and 2.50 kV (−), sheath gas 40 arb and auxiliary gas 10 arb, capillary temperature 325 °C. The first-level (I) and second-level (II) modes parameters were as follows: full scanning resolution 60,000 (I) and 15,000 (II), automatic gain control (AGC) value 1 × 10^6^ (I) and 5 × 10^4^ (II), acquisition rates 12 Hz (I) and 30 Hz (II), the maximum injection times 50 ms (I) and 22 ms (II), ion scanning range (m/z) 100~1000 (I), collision energy of HCD 30% (II). Simultaneous dynamic exclusion was used to remove unnecessary MS/MS information [24]. Data correction was achieved based on the LOESS signal-correction method using pooled QC samples (a mixture of equal volumes from all samples) to eliminate systematic errors. In data quality control, compounds with RSD > 30% in QC samples were filtered out.

### 2.5. HS-GC-IMS Detection of the Volatile Compounds

After the FAA tea samples were brewed with boiling water at the ratio of 1:50, 2 mL of tea broth samples were taken out and placed in a 20 mL headspace flask, and the samples were warmed at 80 °C for 15 min and then injected into the sample, and then, the volatile organic compositions were detected by using a flavor analyzer of the HS-GC-IMS system (Flavour Spec^®^, G.A.S., Dortmund, Germany).

Nitrogen was used as the carrier gas in the experiments, and the compounds were separated on an MXT-5 (15 m, 0.53 mm, 1.0 μm, RESTEK, Bellefonte, PA, USA) column at a column temperature of 60 °C. The carrier gas flow rate was set as follows: 0–2 min: 2.0 mL/min, 2–10 min: 2.0 mL/min–10.0 mL/min, 10–20 min: 10.0 mL/min–100.0 mL/min, and 20–30 min: 10.0 mL/min–150.0 mL/min. A tritium source (3H) was used as the ionization source, which was detected in the positive ion mode with an operating temperature of 45 °C.

### 2.6. HS-SPME/GC×GC-TOFMS Detection of the Volatile Compounds

Samples (100 mg) were mixed with 1 mL of 30% NaCl (saturated solution) and 10 μL of 1 mg/L n-Hexyl-d13 (used as the internal standard). A solid-phase microextraction (SPME) holder installed with a 50/30 μm fiber (DVB/CAR/PDMS, model 57329-U, Supelco, Bellefonte, PA, USA) was used for sampling. Before the adsorption, the fiber was conditioned at 270 °C for 10 min. The SPME extractor was transferred to the incubation chamber, and the samples were adsorbed at 60 °C for 40 min. After the adsorption, the SPME extractor was transferred to the GC inlet and desorbed at 250 °C for 5 min. GC×GC detection was performed using a LECO Pegasus^®^ 4D instrument (LECO, St. Joseph, MI, USA) consisting of an Agilent 8890A GC (Agilent Technologies, Palo Alto, CA, USA) system equipped with a split/splitless injector and dual stage cryogenic modulator (LECO) outfitted with a TOFMS detector (LECO). The separation system consisted of a one-dimensional column, Rxi-5Sil MS (30 m × 250 μm × 0.25 μm) (Restek, Bellefonte, PA, USA), and a two-dimensional column, Rxi-17Sil MS (2 m × 150 μm × 0.15 μm) (Restek, Bellefonte, PA, USA). High-purity helium was used as a carrier gas with a constant flow rate of 1.0 mL/min. For the one-dimensional column Rxi-5Sil MS (30 m × 250 μm× 0.25 μm), the initial temperature was 50 °C for 5 min, then increased to 170 °C at 3 °C/min for 1 min, then increased to 230 °C at 10 °C/min for 5 min. The temperature rise of the 2-D column Rxi-17Sil MS (2 m × 150 μm × 0.15 μm) was 5 °C higher than that of the 1-D column, and the temperature of the modulator was always 15 °C higher than that of the 2-D column. The modulation period was 10 s. The inlet temperature was 250 °C [25]. The mass spectrometric detection of compounds was carried out on a LECO Pegasus BT 4D, with a mass spectrum transmission line temperature of 250 °C, ion source temperature of 250 °C, acquisition rate of 200 spectra/s, electron bombardment source of 70 eV, and detector voltage of 2042 V. The MS scan range was m/z 35–550 [26]. The relative quantity of these compounds was computed based on the concentration of the internal standard. Regarding qualitative analysis, the retention index (RI) was ascertained from the retention times of a series of n-alkanes (C7−C30) through linear interpolation. Volatile organic compounds in samples were identified relying on RIs and comparatively analyzed in conjunction with the recorded mass spectra of each compound by using a mass spectrum library search (NIST).

### 2.7. ROAV Analysis

The ROAV can be used to assess the contribution of each volatile ingredient to the overall flavor [27]. The ROAV is determined using the following equation [28]:ROAV = [(Peak_B_ × T_B_)/(Peak_A_ × T_A_)] × 100(3)
where T_A_ and Peak_A_ are the odor threshold and peak area of the compound with the smallest odor threshold. The standard value of ROAV is set to 100 for component A. T_B_ and Peak_B_ are the odor threshold and peak area of the compound to be tested, respectively.

### 2.8. Data Processing and Analysis

To ensure the reliability of the experimental results, the test was repeated three times and the results were expressed as the mean ± standard deviation (SD). Differences between treatments were analyzed via one-way analysis of variance (ANOVA) at the 0.05 level of significance using IBM SPSS Statistics 27 software. Compound identification was performed using HMDB (https://www.hmdb.ca/, accessed on 10 December 2024, MassBank (https://massbank.eu/MassBank/, accessed on 10 December 2024), McCloud (https://www.mzcloud.org/, accessed on 10 December 2024), and self-constructed substance libraries. The NIST2020 database was used to annotate the raw data with flavor compounds using Chroma TOF library search software (version 5.55.35). The preprocessing steps included normalization using the LOESS signal correction method based on pooled QC samples to eliminate systematic errors, log2 transformation to reduce the influence of high-intensity peaks and improve data distribution, and autoscaling (mean-centered and divided by the standard deviation) to ensure comparability across variables. PCA, PLS-DA, and OPLS-DA were performed using R (https://www.r-project.org, accessed on 10 December 2024) and SIMCA-P (v13.0, Umetrics, Umea, Sweden), respectively. Based on the Kyoto Encyclopedia of Genes and Genomes (KEGG, https://www.genome.jp/kegg/, accessed on 10 December 2024), metabolic pathway enrichment analysis was performed. Histograms, clustered heatmaps, and volcano maps were drawn using R software (plot package and heatmap package, version 1.22.0).

## 3. Results and Discussion

### 3.1. Appearance Properties of FAA Tea at Four Harvest Times

As illustrated in Figure 1a, FAA tea produced from March to June exhibited significant morphological and color changes. The color shifted from yellow to dark green, and the tea extracts became visibly darker. These changes can be attributed to the high content of chlorophyll, carotenoids, flavonoids, and other pigmented compounds in FAA tea, which inevitably affect the color of the product [29].

The color of the FAA tea soup plays a crucial role in consumer acceptance. Results from the color difference analysis, shown in Figure 1b–d, revealed variations in the L*, a*, and b* values across the four harvest periods. Here, L* represents brightness (0 = black, 100 = white), a* denotes the red–green axis (+a = red, −a = green), and b* indicates the yellow–blue axis (+b = yellow, −b=blue) [30]. Over the harvest periods, the brightness (L*) initially decreased and then increased, while both a* and b* values initially rose and subsequently declined. FAA-5 marked a turning point, with the lowest L* and highest a* and b* values, corresponding to the darkest color. In contrast, FAA-6 exhibited intermediate L*, a*, and b* values, resulting in a bright and visually appealing tea soup.

### 3.2. Analysis of Antioxidant Capacity of FAA Tea at Four Harvest Times

Free-radical-scavenging activity is usually expressed as the percentage of free radicals neutralized, but it can also be quantified using the IC_50_ value, which represents the concentration of antioxidants required to remove 50% of free radicals [31]. A lower IC_50_ value indicates a stronger scavenging ability of free radicals [32]. As shown in Figure 1e–g, the antioxidant activity of FAA tea, measured through DPPH•, ABTS•^+^, and FRAP assays, progressively increased from March to June. The growth rate of antioxidant capacity was more pronounced from March to May, while it slowed in June. Similarly, the IC_50_ values for DPPH• and ABTS•^+^ exhibited a downward trend with increasing harvest periods, indicating enhanced antioxidant activity. However, the rate of decrease in IC_50_ values became less pronounced between May and June. Overall, the antioxidant activity of FAA tea improved with later harvest dates, showing rapid increases in earlier harvests and slower gains from May to June. This pattern aligns with the findings of Mi et al. [33], further validating the relationship between harvest timing and antioxidant properties in plant-based products.

### 3.3. Characterization of Non-Volatile Metabolites of FAA Tea at Four Harvest Times

#### 3.3.1. Comprehensive Analysis of Non-Volatile Metabolites in FAA Tea

A total of 45,982 peaks, comprising 23,407 negative ion peaks and 22,575 positive ion peaks, were detected in FAA tea across four different harvest periods. Specifically, a total of four sample groups with three biological replicates were explored in this study. Ion peaks with inaccurate characterizations were excluded based on reference thresholds, resulting in the detection of 8627 metabolites, including 5009 in positive mode and 3618 in negative mode. Of these, 509 compounds were identified at the secondary level.

#### 3.3.2. Multivariate Statistical Analysis of Non-Volatile Metabolites

To systematically explore the metabolic differences among FAA tea samples harvested at different periods, multivariate statistical analyses were performed, including principal component analysis (PCA), partial least squares discriminant analysis (PLS-DA), and orthogonal partial least squares discriminant analysis (OPLS-DA) [34,35]. All analyses were conducted using the R package Ropls, with data preprocessed through mean-centering and unit variance scaling.

The PCA multivariate statistical analysis of metabolites was performed to compare the metabolite peaks among the four sample groups and the QC group (Figure 2a,b). The first principal component (PC1) explained 31.8% and 30.7% of the variance in positive and negative modes, respectively, while the second principal component (PC2) accounted for 15.2% and 13.3% of the variance. The PCA results revealed significant segregation of metabolites during different harvest periods (PC1), with greater PC1 values indicating larger metabolomic differences among sample groups. The PC2 results indicated changes in FAA tea metabolism during its developmental stages.

To enhance class separation and validate multiclass discrimination, PLS-DA was employed. In the positive mode, the first principal component (PC1) captured 30.2% of the variance, and in the negative mode, it explained 29%. Meanwhile, the second principal component (PC2) represented 15.4% and 13.9% of the variance in the positive and negative modes, respectively (Appendix A). The model demonstrated high explanatory power, with R2Y = 0.99 and Q2 = 0.917 in positive ion mode and R2Y = 0.992, Q2 = 0.896 in negative ion mode. Permutation tests (200 iterations) confirmed that the model could fit and predict the data quite well (Appendix A).

For pairwise comparisons, OPLS-DA models were constructed to identify key discriminative metabolites [36]. The Q2(cum) of the OPLS-DA model was 0.99(+) and 0.992(−), respectively, which indicated that the model a good predictive performance. Then, 7-fold cross-validation and 200 response alignment tests were performed, and the results showed that the model was not overfitted (Figure 2c,d). Hierarchical cluster analysis indicated that the FAA-4 group and FAA-5 group had the closest distance and were clustered together, while the FAA-6 group had the farthest relationship with the other three groups. The qualitative difference in metabolites between different harvest periods of FAA tea was illustrated (Figure 2e,f).

The integration of PCA, PLS-DA, and OPLS-DA provided complementary insights. PCA revealed that the harvest time was the dominant source of variation (PC1), and PLS-DA confirmed multiclass separability. OPLS-DA-driven pairwise comparisons pinpointed FAA-6 as the most metabolically distinct group, which was consistent with its unique differentially accumulated metabolite (DAM) profile. Moreover, the minimal differences between FAA-4 and FAA-5 were in line with their clustering in HCA, suggesting a transitional metabolic state during these periods.

#### 3.3.3. Characterization of Differential Non-Volatile Metabolites in FAA Tea

Metabolite accumulation is a complex and important process in plants, influenced by genetic and environmental factors [33]. In this study, UPLC-MS/MS analysis detected a large number of metabolite profiles belonging to different classes, offering valuable insights for discovering new compounds related to the flavor and quality control of FAA tea.

DAMs were identified through a two-step strategy: first, pairwise OPLS-DA comparisons were carried out, where for each harvest time pair (such as FAA-3 vs. FAA-4), metabolites with VIP > 1 and *p* < 0.05 (*t*-test) were selected; second, cross-group integration was performed, with metabolites that were consistently altered across multiple comparisons being prioritized as candidate markers. Differentially accumulated metabolites were identified using a *t*-test with screening criteria of VIP >1 and *p* < 0.05 [37]. The volcano plots (Figure 3c–h) explained the up-and-down-regulation information of various metabolites among FAA tea samples at different harvest periods. The metabolite changes of FAA-3 vs. FAA-4, FAA-3 vs. FAA-5, FAA-3 vs. FAA-6, FAA-4 vs. FAA-5, and FAA-5 vs. FAA-6 were 146 (76 up/70 down), 144 (54 up/90 down), 216 (89 up/127 down), 92 (23 up/69 down), 214 (88 up/126 down), and 207 (98 up/109 down). These results highlight significant differences in metabolic profiles across developmental stages of FAA tea. To differentiate FAA tea samples, an upset plot (Figure 3a) identified 13 unique metabolites in FAA-6 and FAA-3, 11 unique differential metabolites in the comparison group of FAA-6 and FAA-5, and 12 unique differential metabolites in the comparison group of FAA-6 and FAA-4. At the super classification level, differential metabolites were grouped into 11 categories, including lipids and lipid-like molecules (29.07%), organic acids and derivatives (18.66%), benzenoids (12.97%), phenylpropanoids and polyketides (11.20%), organoheterocyclic compounds (10.41%), and organic oxygen compounds (8.64%) (Figure 3b). Metabolite abundances varied significantly across harvest periods, demonstrating a strong influence of the developmental stage [38].

A total of 154 differential metabolites (Appendix A) were screened out through the comparison of four groups of samples, with five significant marker metabolites identified: a compound tentatively identified as kynurenic acid, 5-methoxyindoleacetate, 4-quinolinecarboxylic acid, gamma-aminobutyric acid (GABA), and fructose-1P. As illustrated in Appendix A, which shows the characterization of the chemical structures of key compounds in FAA tea harvested at four different times, the structural features of these metabolites can provide insights into their potential functions. Kynurenic acid is an antagonist of excitatory amino acid receptors that also reduces quinolinic-acid-mediated neurotoxicity, obtained by metabolizing L-tryptophan [39]. The kynurenic acid in FAA-6 was the highest, which was 2.5 times that in FAA-3, the lowest. Fructose-1P, contributing to the sweetness of FAA tea, also showed distinct variations across the harvest periods. These findings highlight the dynamic metabolic changes in FAA tea across harvest times and invite further analysis of their biological implications.

The dynamic metabolic profile revealed by our study, particularly the tentative identification of kynurenic acid as a crucial differential metabolite, differs from previous reports on FAA, which primarily focused on flavonoids (e.g., quercetin, kaempferol) and volatile oils (e.g., eucalyptol) as key components [40]. This discrepancy may be attributed to several factors, including variations in growing conditions, extraction methods (HS-GC-IMS, HS-SPME/GC×GC-TOFMS, and UPLC-MS/MS), or metabolic regulation [41,42]. Notably, our non-targeted metabolomics approach identified 154 non-volatile differential metabolites, among which kynurenic acid, 5-methoxyindoleacetate, 4-quinolinecarboxylic acid, fructose-1P, and gamma-aminobutyric acid were the most significant.

It is important to note that key substances and key differential substances are not identical. Key substances typically refer to compounds with high abundance or significant bioactivity, while key differential substances are those that show the most significant changes under different conditions or treatments. For example, previous studies have highlighted the antioxidant and anti-inflammatory activities of flavonoids and polysaccharides in FAA, which are considered key substances due to their biological significance [40]. In contrast, our study focused on differential metabolites that vary significantly across harvest times, providing new insights into the dynamic metabolic profile of FAA tea.

The potential presence of kynurenic acid in FAA is intriguing given its known biological activities, such as neuroprotection and antioxidant effects [43,44]. However, as the identification of kynurenic acid is currently based on LC-MS/MS data and multivariate statistical analysis, further confirmation through co-elution experiments and NMR analysis is required in future studies [42]. This limitation highlights the need for more comprehensive structural verification to fully validate our findings.

#### 3.3.4. Kyoto Encyclopedia of Genes and Genomes (KEGG) Analysis of Differential Metabolites

To investigate the potential pathways of the effect of the harvest period on FAA tea metabolites, functional enrichment analysis was performed using the KEGG database (Figure 4). A total of 154 differentially accumulated metabolites were mapped to 30 metabolic pathways. Among them, the biosynthesis of plant secondary metabolites exhibited the most significant impact, encompassing 20 differentially accumulated metabolites, including L-glutamic acid, succinic acid, L-arginine, L-glutamine, L-phenylalanine, L-tyrosine; L-malic acid; L-asparagine; citric acid; phenylpyruvic acid; L-valine; palmitic acid; nicotinic acid, prephenate; geranyl diphosphate; L-dopa; 4-hydroxyphenylpyruvic acid; campesterol; stigmasterol; and (S)-abscisic acid. Chen et al. highlighted that amino acids, such as glutamic acid, aspartic acid, glutamine, and asparagine, contribute to fresh flavor, with glutamic acid being a key determinant of taste [45]. The enrichment of these metabolites underscores their potential role in shaping the flavor profile of FAA tea. Metabolic pathway analysis revealed the complex biological activities involved in metabolite accumulation and highlighted significant differences between FAA-3 and FAA-6. These differences were primarily associated with amino acid biosynthesis pathways, suggesting that the harvest period significantly affects metabolite pathways. In contrast, metabolite differences between FAA-4 and FAA-5 were minimal, consistent with the PCA and HCA results, which showed closer clustering of these groups. This indicates that intermediate harvest periods have less pronounced effects on the metabolite composition.

### 3.4. Characterization of Volatile Compounds of FAA Tea in Four Harvest Times

#### 3.4.1. Analysis of Volatile Compounds by HS-GC-IMS

HS-GC-IMS is distinguished by its simplicity, speed, and high sensitivity. Figure 5a,b show the three-dimensional and two-dimensional spectra of FAA tea, illustrating the differences in volatile compounds across various harvest periods. To compare this difference more obviously, the spectrum of FAA-3 was selected as a reference, with spectra from other samples subtracted from it. In this differential visualization, white indicates no difference, red signifies a higher concentration of a compound relative to the reference, and blue indicates a lower concentration. The two-dimensional difference spectrum (Figure 5c) clearly shows significant variations in volatile compounds between FAA tea samples from different harvest periods. To further investigate, all peaks were selected for fingerprint analysis.

Based on the identified compounds, PCA analysis was performed to explore the diversity among FAA samples, as shown in Figure 6a. The principal components PC1 and PC2 accounted for 51% and 18% of the total variance, respectively. Therefore, the distinct separation of FAA-6 from other FAA samples based on PC1 and PC2 reflects the significant differences among them. As can be seen in Figure 6b, the fingerprints of volatile flavor compounds in the FAA tea samples identified 44 known flavor compounds, including 20 aldehydes, 6 alkenes, 9 alcohols, 6 ketones, 2 ethers, and 1 ester. Aldehydes and alcohols were significantly higher in FAA-3 than in other harvest periods, while ketones peaked in FAA-5, and alkenes, ethers, and esters were most prevalent in FAA-6. Specific compounds like benzaldehyde, with its special almond aroma and (Z)-4-heptanal with grassy aroma, were more prominent in the early harvest period. With the prolongation of the harvest period, compounds, such as 6-methyl-5-hepten-2-one, pentanol, and nonanal, increased and then decreased, peaking during the middle to late harvest stages. Certain compounds, including hexanal, 2,3-butanedione, linalool, rose ether, α-terpinodiene, γ-terpinene, β-pinene, heptanal, α-pinene, (E)-2-hexenol, (E)-2-pentenal, penten-3-one, penten-3-ol, octanal, α-salicylic acid alkene, gibberellic acid, camphor, gibberellin acetate, and 1,8-eucalyptus brain, showed a gradual increase in content, with high levels observed during the late harvest stage. Others, such as (Z)-2-heptenal and pentanal, associated with fruity and bready aromas, were more abundant in the early and mid-late harvest periods. These findings highlight the dynamic changes in FAA tea’s volatile flavor compounds across harvest stages, directly influencing its aromatic profile.

#### 3.4.2. Analysis of Volatile Compounds via HS-SPME/GC×GC-TOFMS

The chromatograms obtained from HS-SPME/GC×GC-TOFMS analysis are shown in Appendix A. The chromatograms of FAA tea harvested at different times vary. Flavor compounds in FAA tea consist primarily of volatile compounds, such as hydrocarbons, aldehydes, esters, acids, ketones, alcohols, ethers, phenols, and heterocyclic compounds, which are generated through complex biochemical reactions involving flavor precursors during processing [46]. As shown in Figure 7a, FAA-6 exhibited the highest number of volatile compound species, while FAA-3 had the lowest. Figure 7b highlights that a total of 1742 volatile compounds were identified, with 443 shared across all four harvest periods. Among the unique compounds, FAA-3 contained 159, FAA-4 had 150, FAA-5 included 173, and FAA-6 had the highest number at 298. Figure 7c further illustrate variations in the relative content of volatile compounds across the four harvest periods. Notably, FAA-6 showed the highest abundance and relative content of ketones and esters, emphasizing its distinct flavor profile compared to earlier harvest periods. These results demonstrate that the harvest period significantly impacts the composition and relative abundance of flavor compounds in FAA tea.

#### 3.4.3. Key Flavor Characteristics of FAA Tea

At present, the relative odor activity value (ROAV) is commonly used to evaluate the contribution of volatile compounds to the overall flavor. When the ROAV of a compound is >1, it is considered a key flavor compound. Figure 7d illustrates the variation in flavor compound contributions among FAA tea samples from different harvest periods. In this study, 36 volatile compounds with ROAV > 1 in FAA tea were identified across all samples (Appendix A). FAA-6 had the highest number of key flavor compounds (34), while FAA-3 contained the least (20). A total of 18 volatile compounds with ROAV > 1 were identified in FAA-3, FAA-4, FAA-5, and FAA-6. Among these, 12 compounds exhibited their highest ROAVs in FAA-6, ranging from 2 to 43 times the levels observed in other harvest periods. Additionally, 13 unique compounds with ROAV > 1 were exclusive to FAA-6, indicating its richer aromatic composition. The compound with the highest ROAV among all FAA tea samples was 1-octen-3-one, followed by (E)-2-octenal, isovaleric acid, (E)-2-undecenal, heptanal, and (E)-2-nonenal. The 1-octen-3-one with the highest ROAV has strong soil, mushroom, and metallic aromas with vegetable notes, like cabbage and cauliflower, and (E)-2-octenal with the second highest ROAV has fat and meat notes, as well as cucumber and chicken-like notes. Isovaleric acid and heptanal give off a sweet and fruity aroma, as well as a Duke’s lingonberry-like flavor. (E)-2-Undecenal has a strong fresh aldehyde odor. And (E)-2-nonenal has a sweet or cucumber flavor [47].

The flavor of a product consists of recognizable taste and smell properties, and a complex of properties cannot be identified separately. FlavorDB was used to analyze and compare the sensory flavors of compounds [48]. As shown in the Figure 7e radar map for the sensory flavor characteristic analysis, a sweet aroma among the characteristic aromas of the four FAA tea samples was the most prominent in all of them, followed by a green aroma and fruity aroma. The FAA tea from different harvest periods differed in other aromas, except for no significant difference in the bitter aroma. The woody, floral, and clean aroma of FAA-3 was more prominent, while the sweet, green, fruity, herbal, and waxy aroma of FAA-6 was more prominent, but the waxy and medicinal aroma of FAA-4 was the weakest, and the sweet, herbal, fruity, and floral aroma of FAA-5 was weaker.

A principal component analysis (PCA) of FAA tea samples from different harvest periods is shown in Figure 8a, where PC1 explained 29% of the total variance, while PC2 explained 17.2% of the total variance, with a combined contribution of 46.2%. The proximity between the three replicates in each group indicates the reliability of the results. The four sample groups were clearly separated, suggesting that the volatile principal components of the FAA tea samples differed among the different harvest periods. FAA-3, FAA-4, and FAA-5 were clustered on one side, far away from FAA-6, suggesting that the volatile components of FAA-6 differed most from those of the other three FAA tea harvest periods. In order to better compare the differences in FAA tea aromas among different harvest periods, the OPLS-DA model was established based on the relative contents of volatile compounds of FAA tea species. The R2Y and Q2 of the models were 0.995 and 0.877, respectively, indicating that they were effective in distinguishing FAA tea samples from different harvest periods. The cross-validation test (Figure 8b) showed that the results were accurate and there was no overfitting of the models.

As described in Section 3.3.3, first, pairwise OPLS–DA comparisons were conducted for each harvest time pair, and metabolites with VIP > 1 and *p* < 0.05 (*t*-test) were chosen. Second, cross-group integration was carried out, prioritizing metabolites consistently altered across multiple comparisons as DAMs. And 75 differential volatile compounds were screened. A heatmap was drawn to visualize these differential volatiles, showing the variation among harvest times. As illustrated in Figure 8c, the compound abundances of FAA-3 and FAA-6 showed the greatest difference, while the abundances of FAA-4 and FAA-5 were closer. This is consistent with the PCA results, indicating that these differential compounds are the key to the flavor differences among FAA tea in different harvest periods. In contrast, FAA-3, FAA-4, and FAA-5 had higher contents of camphor, caryophyllene, lavandulol, and benzeneacetaldehyde, which contributed more to their woody, floral, and sweet aromas. In FAA-6, artemisia alcohol, (−)-carvone, pinocarvone, and (+)-3-thujone were volatile compounds with an “herbal/sweet/fresh/green” aroma. In addition to the differential metabolites discussed above, α-thujene was also identified as a significant volatile compound in the HS-SPME/GC×GC-TOFMS analysis (Figure 8c), although it was not detected in the HS-GC-IMS analysis. This discrepancy may be attributed to the differences in sensitivity and selectivity between the two analytical techniques. α-Thujene is a well-known component of Artemisia essential oils and has been reported to exhibit various biological activities, including antimicrobial and anti-inflammatory properties [49]. Its presence in our HS-SPME/GC×GC-TOFMS data further supports the complexity and diversity of volatile compounds in Artemisia species. Future studies could explore the specific roles of α-thujene in the biological activities of Artemisia.

The sensory flavor characteristics and flavor compound network diagram of FAA tea were constructed based on lgraph and Flavordb. As seen in Figure 8d, we selected 10 frequent flavors and 29 related differential compounds for the graph. The three differential compounds with ROAV ≥ 1 are the key flavor compounds: 1-hexanol, benzeneacetaldehyde, and 2-undecanone. The content and ROAV of 1-hexanol in FAA-6 were both the highest, which was 10 times that in the lowest FAA-3, and contributed a strong green aroma to FAA-6. The high content of benzeneacetaldehyde in FAA-4 provided it with floral and honey aromas. The odor threshold of 2-undecanone was the lowest, and the ROAV value in FAA-6 was relatively high, emitting orange, fresh, and green aromas. In terms of the flavor composition, the sweet aroma contributed the most to the FAA tea flavor, followed by green, woody, and floral aromas. Fifteen volatile compounds were contributing to the sweet aroma, and nine compounds were related to the green. The woody and floral aromas were also formed by the six volatile compounds, respectively.

### 3.5. Correlation Analyses of Antioxidant Power, Key Flavor Compounds, and Differential Non-Volatile Metabolites

Correlation analyses were performed to investigate the relationships among antioxidant capacity, key flavor compounds, and non-volatile differential metabolites in FAA tea. As shown in Figure 9, compounds, such as D-xylitol, L-glutamic acid, honokiol, and costunolide, exhibited a highly positive correlation (r = 1) with the antioxidant activity of FAA tea. These findings suggest that these compounds play pivotal roles in the antioxidant processes of FAA tea. Previous studies indicate that these compounds may synergistically enhance the antioxidant capacity through distinct mechanisms, thus protecting organisms from oxidative damage [50,51,52]. Specifically, honokiol has been confirmed to be a strong oxidant [53]. D-xylitol, L-glutamic acid, and costunolide might indirectly contribute to the antioxidant properties of FAA tea by participating in the regulation of the synthesis of antioxidant compounds in the body and stabilizing the microenvironment.

The proportion of non-volatile compounds in FAA tea during different harvest periods could affect the perception of aroma. Non-volatile compounds often participate in metabolic pathways and metabolic reactions, resulting in the formation of volatile metabolites. After summarizing the influences of different harvest periods on the volatile and non-volatile metabolites of FAA tea, it was necessary to explore the correlation between non-volatile and volatile metabolites in FAA tea at different harvest periods. It can be seen in Figure 9 that there were different situations of strong positive correlations, strong negative correlations, and weak correlations between 30 key flavor compounds and 32 differential metabolites. Among them, the correlation between esters and ketones among flavor compounds and lipids and organic acids among metabolites was generally strong. This meant that the presence of certain compounds might enhance the flavor of FAA tea, while others might have an inhibitory effect on the flavor. Therefore, the difference in harvest periods affected the intricate biochemical reactions in FAA tea and played a crucial role in flavor formation. To our knowledge, there are currently no literature reports on the key flavor compounds and non-volatile metabolites in FAA tea at different harvest periods and their correlations. A better understanding of this topic would help enhance the key flavor components and produce the desired products by controlling the harvest period of FAA tea or regulating non-volatile metabolites.

## 4. Conclusions

A comprehensive metabolomic analysis of FAA tea demonstrates that the harvest time significantly influences its flavor and antioxidant properties. Volatile compounds, including olefins, ethers, and esters, reached their highest levels in June, with key flavor compounds, such as 1-octen-3-one and (E)-2-octenal, contributing most to the tea’s distinctive flavor profile. Non-volatile metabolites also showed marked differences across harvest periods, with kynurenic acid, gamma-aminobutyric acid, and other metabolites serving as potential biomarkers. The antioxidant capacity of FAA tea increased with the length of the harvest period, peaking in June, and was strongly correlated with D-xylitol, L-glutamic acid, honokiol, and costunolide. These results emphasize the critical role of harvest timing in optimizing both the flavor quality and antioxidant potential, providing valuable insights for improving production standards in the FAA tea industry.

## Figures and Tables

**Figure 1 foods-14-00843-f001:**
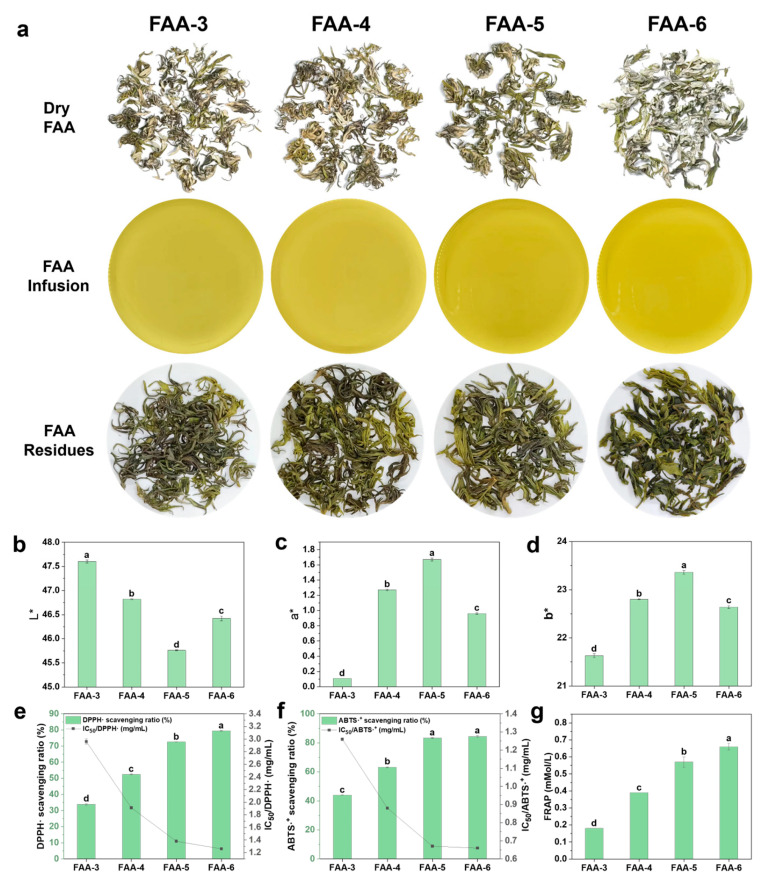
The FAA tea appearance characteristics. (**a**) Image of dry FAA tea, FAA tea infusions, and FAA tea residues; (**b**) L* value; (**c**) a* value; (**d**) b* value; (**e**) the DPPH• scavenging ratio (%) and IC50/DPPH• (mg/mL); (**f**) ABTS•+ scavenging ratio (%) and IC50/ABTS•+ (mg/mL); (**g**) FRAP (mmol/L). (The letters in the figures represent the significant differences among different treatments.)

**Figure 2 foods-14-00843-f002:**
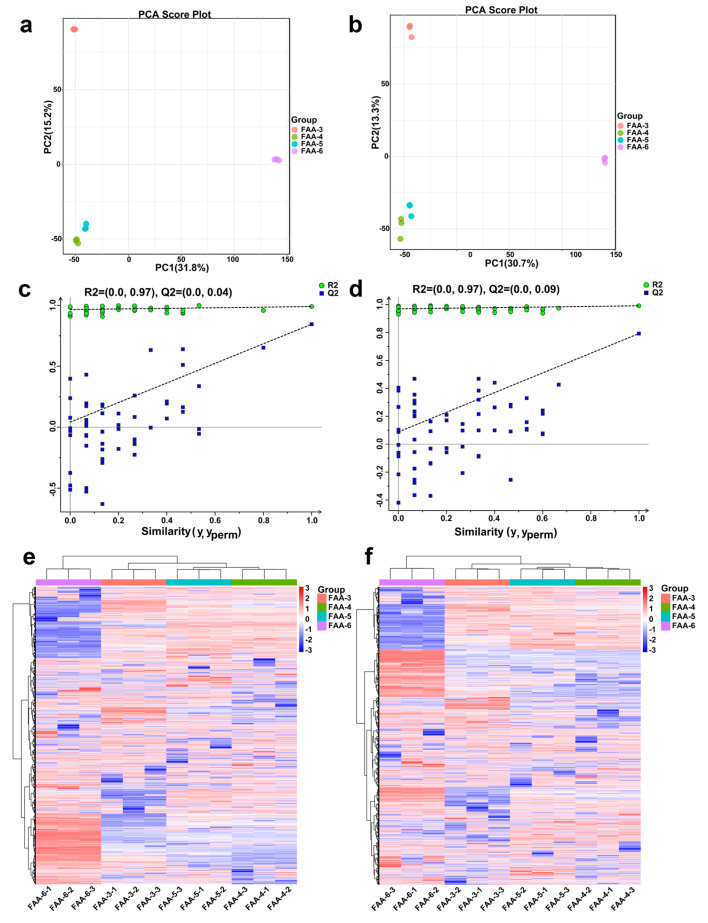
(**a**,**b**) Metabolome principal component analysis (PCA) of 12 FAA samples and quality control samples (QC) in positive (+) and negative (−) ion modes; (**c**,**d**) results of PLS-DA arrangement tests in positive (+) and negative (−) ion modes; (**e**,**f**) the heat map of cluster analysis in positive ion (+) and negative ion (−) modes, respectively.

**Figure 3 foods-14-00843-f003:**
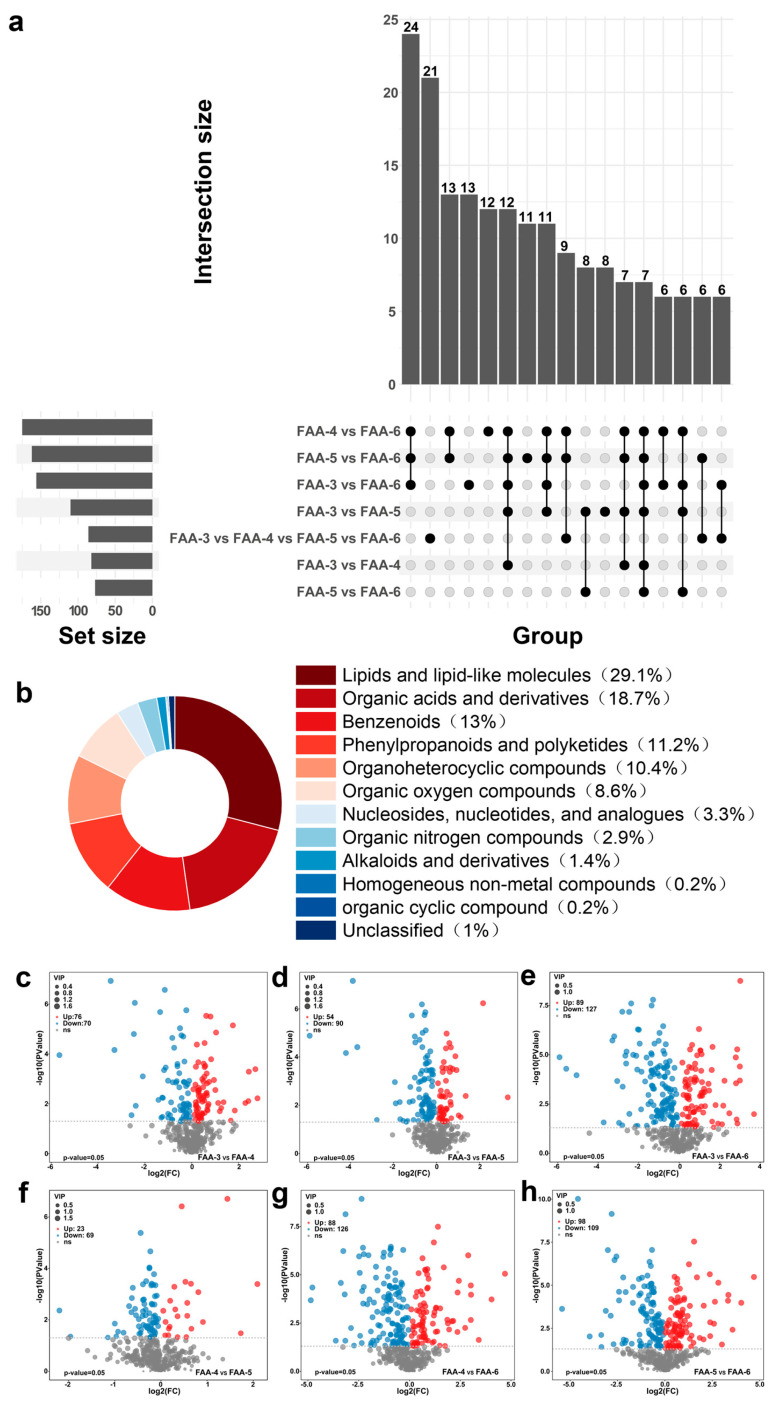
(**a**) Upset plot of differentially accumulated metabolites (DAMs) in four FAA tea samples; (**b**) classification loops of identified differential metabolites at the superclass level; (**c**–**h**) volcano plot of FAA tea samples.

**Figure 4 foods-14-00843-f004:**
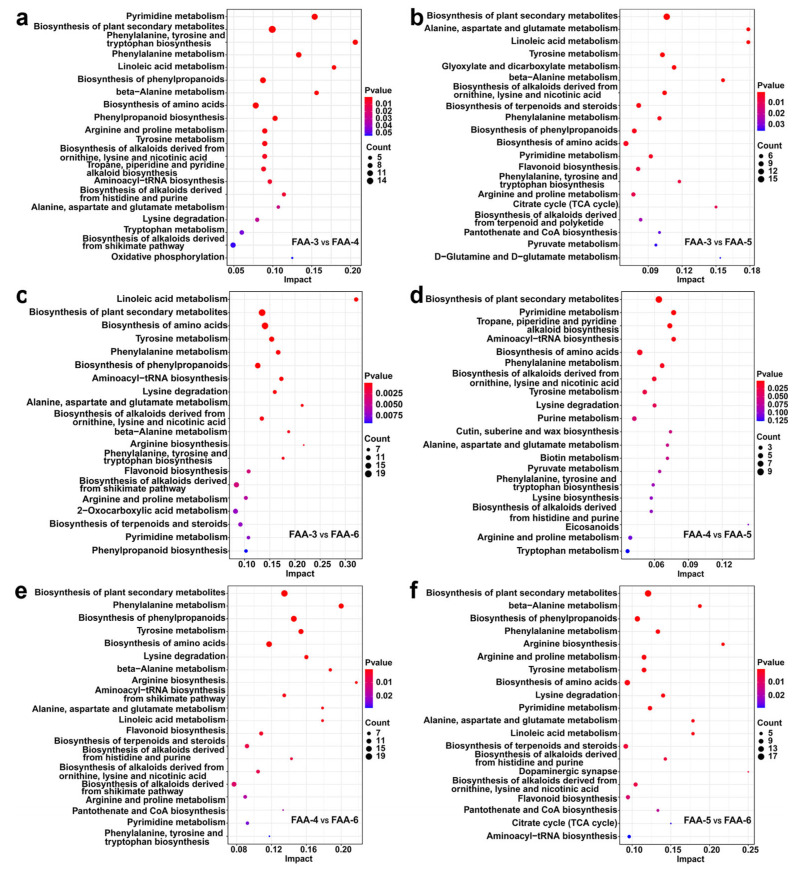
(**a**–**f**) KEGG enrichment bubble chart (FAA-3 vs. FAA-4, FAA-3 vs. FAA-5, FAA-3 vs. FAA-6, FAA-4 vs. FAA-5, FAA-4 vs. FAA-6, FAA-5 vs. FAA-6).

**Figure 5 foods-14-00843-f005:**
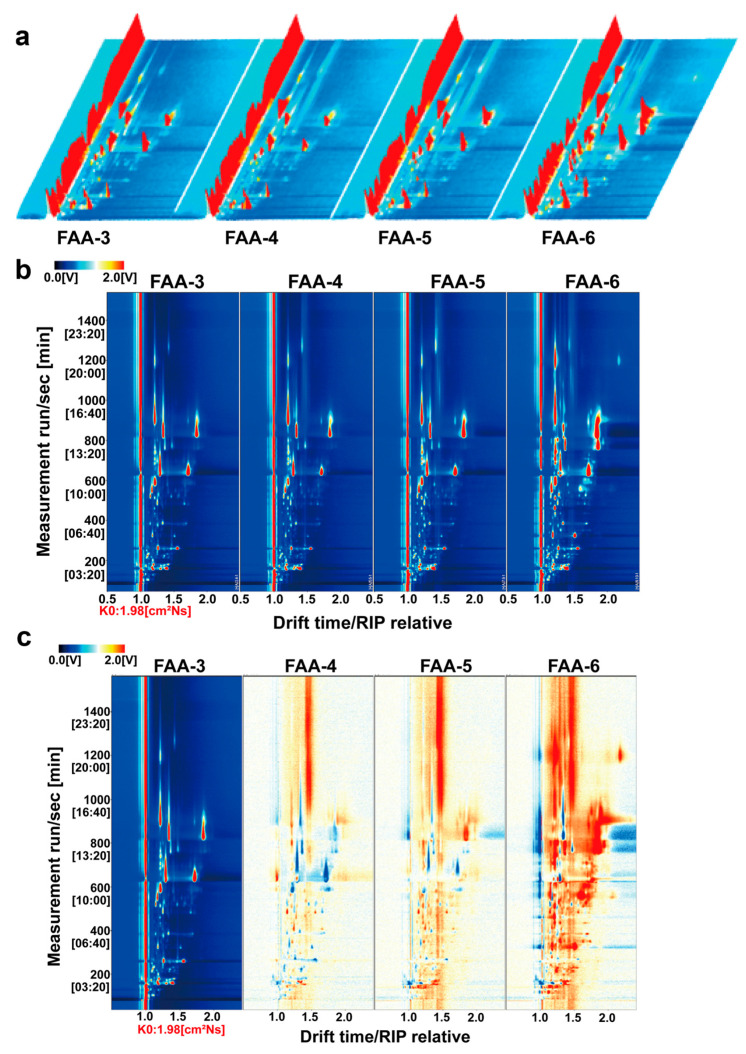
GC-IMS analysis of volatile components of FAA tea under different harvest times. (**a**) Three-dimensional HS-GC-IMS spectra of volatile organic compounds in FAA tea samples; (**b**) HS-GC-IMS two-dimensional spectra of volatile organic compounds in FAA tea samples; (**c**) differential HS-GC-IMS spectra of volatile organic compounds in FAA tea samples.

**Figure 6 foods-14-00843-f006:**
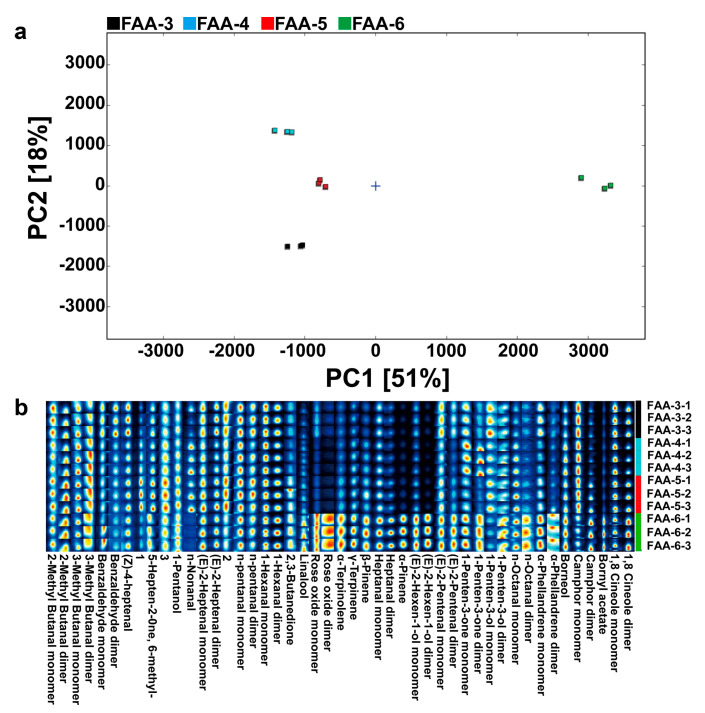
(**a**) PCA score plots based on GC-IMS analysis; (**b**) FAA tea flavor fingerprint via GC-IMS analysis.

**Figure 7 foods-14-00843-f007:**
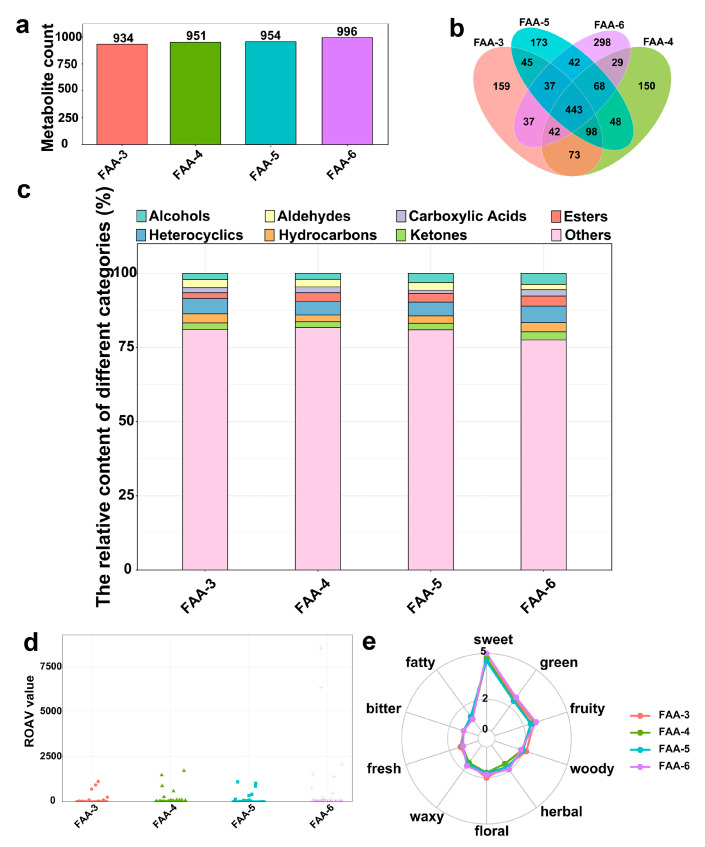
(**a**) Metabolite count of four FAA tea samples; (**b**) Venn chart of group material overlap numbers; (**c**) accumulation diagram of relative content of species; (**d**) scatter plot of ROAV relative odor activity value; (**e**) radar map for analysis of sensory flavor characteristics. (The outermost name represents the sensory flavor characteristics, and the broken line represents the frequency grade of the corresponding flavor compound [the detection frequency is graded from 1–5, the highest frequency is grade 5], and the color indicates different groups.)

**Figure 8 foods-14-00843-f008:**
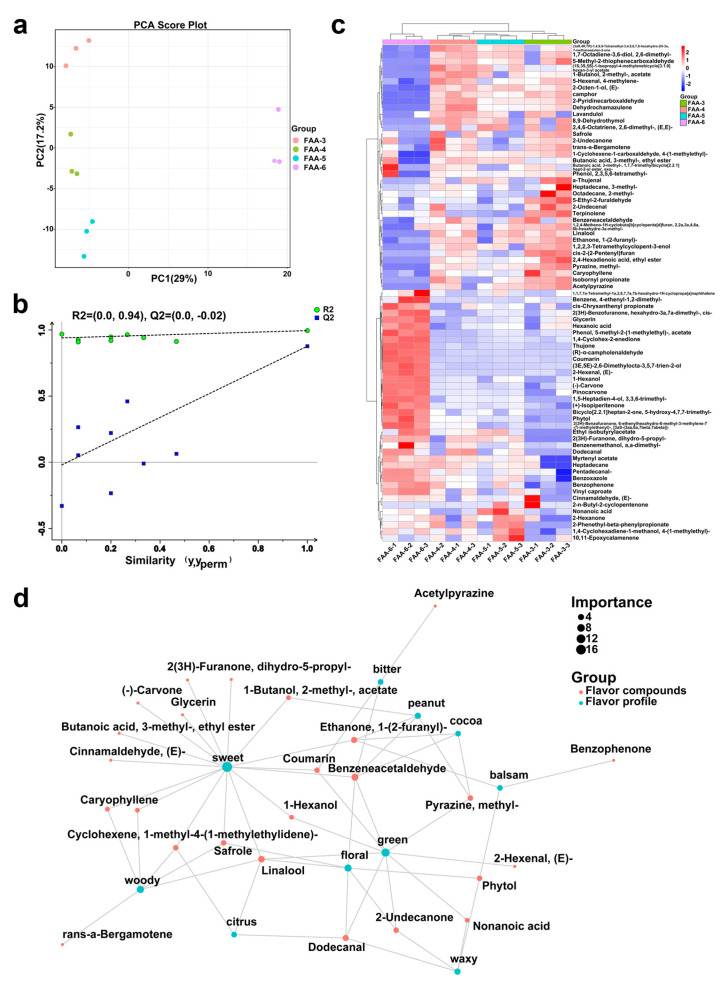
(**a**) PCA score plots; (**b**) OPLS-DA permutation test; (**c**) differential molecular heat map; (**d**) sensory flavor characteristics and flavor compound network.

**Figure 9 foods-14-00843-f009:**
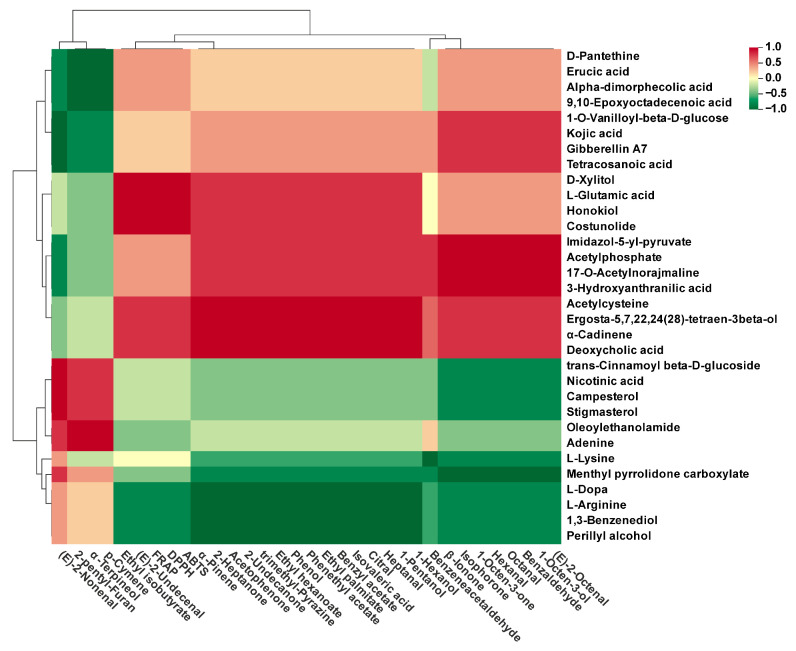
Correlation analysis of antioxidant power, key flavor compounds, and differential non-volatile metabolites.

## Data Availability

The original contributions presented in the study are included in the article/Appendix A, further inquiries can be directed to the corresponding authors.

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
