# Peer review of "A Comprehensive Metabolomic Analysis of Volatile and Non-Volatile Compounds in Folium Artemisia argyi Tea from Different Harvest Times"

_foods, 2025, doi:10.3390/foods14050843_

Round 1
Reviewer 1 Report
Comments and Suggestions for Authors
The paper "A comprehensive metabolomic analysis of volatile and nonvolatile compounds in Folium Artemisia argyi tea from different harvest Times" contains appropriate references to the experiment in the introduction. The authors have the appropriate experience and workshop. In their work, they used universal research methods for determining the antioxidant activity of the subjects and determining the color. In the work, the researchers used chromatography techniques at the highest world level. The metadata obtained is developed in a sufficiently clear and transparent manner.
The authors did not avoid a few shortcomings that will need to be corrected.
Noticed defects:
Questions
• How were the samples stored? Line 72
• DPPH• radical scavenging assay was tea powder actually dissolved in anhydrous ethanol?
Was Tea powder actually dissolved in anhydrous ethanol?
Please tell us how to prepare the Tea powder sample.
• Was the method of sample preparation for ABTS•+ radical scavenging assay and Ferric-reducing antioxidant power (FRAP) assay the same as in the DPPH• radical scavenging assay method? If so, please clearly and without ambiguity indicate in the methodology of ABTS and FRAP marking.
• The average leaf area increased, was this parameter measured, if not, there is no need to write about it, or describe the basis of the description? Line 208
• Figure 6 c and d illustrate the same data, one of the graphs seems to be unnecessary. Page 13
Author Response
Thank you so much for your letter and the comments from the reviewers about our manuscript (foods-3448036). We tried our best to improve the manuscript and made some changes, which will not influence the content and framework of the paper. Here we did not list the changes but marked them in red in the revised paper. The comments from the reviewers are all valuable and very helpful for revising and improving our paper, as well as the important guiding significance to our research. We have studied comments carefully and have made corrections which we hope to meet with approval. Revised portions are marked in red on the paper. The main corrections in the paper and the responses to the reviewer’s comments are as follows:
Response to reviewer 1
Reviewer #1: The paper "A comprehensive metabolomic analysis of volatile and nonvolatile compounds in Folium Artemisia argyi tea from different harvest Times" contains appropriate references to the experiment in the introduction. The authors have the appropriate experience and workshop. In their work, they used universal research methods for determining the antioxidant activity of the subjects and determining the color. In the work, the researchers used chromatography techniques at the highest world level. The metadata obtained is developed in a sufficiently clear and transparent manner.
The authors did not avoid a few shortcomings that will need to be corrected.
Noticed defects:
Comment 1. How were the samples stored? Line 72
Response: Thank you for pointing this out. The FAA samples were stored at -80℃ to maintain their stability and prevent degradation until further analysis. We have updated the manuscript to include this information (Lines 80-81).
Comment 2. DPPH• radical scavenging assay was tea powder actually dissolved in anhydrous ethanol? Was Tea powder actually dissolved in anhydrous ethanol? Please tell us how to prepare the Tea powder sample.
Response: Thank you for your questions. The tea powder was added to anhydrous ethanol but did not completely dissolve. To ensure homogeneity, the mixture was vigorously vortexed before centrifugation. The FAA tea powder was prepared by freeze-drying FAA tea, followed by grinding and sieving through a 100-mesh sieve to obtain a fine powder. We have revised the manuscript to clarify these points, and the changes are marked in red (Lines 94-98).
Comment 3. Was the method of sample preparation for ABTS•+ radical scavenging assay and Ferric-reducing antioxidant power (FRAP) assay the same as in the DPPH• radical scavenging assay method? If so, please clearly and without ambiguity indicate in the methodology of ABTS and FRAP marking.
Response: Thank you for your careful examination. In response, we have revised the manuscript to explicitly state that the sample preparation method for both the ABTS•+ and FRAP assays was consistent with that described in section 2.3.1 for the DPPH• assay. The relevant sections (2.3.2 and 2.3.3) have been updated to eliminate any ambiguity (Lines 108-109, 119-120).
Comment 4. The average leaf area increased, was this parameter measured, if not, there is no need to write about it, or describe the basis of the description? Line 208
Response: Thank you for your comment. We acknowledge that the average leaf area was not measured, and we have removed the related statement from the manuscript (Lines 230-231). The revised text now focuses on the observed color changes, which are supported by the data in Figure 1a. We appreciate your feedback, which has improved the clarity of our manuscript.
Comment 5. Figure 6 c and d illustrate the same data, one of the graphs seems to be unnecessary. Page 13
Response: Thank you for the kind reminder. You're correct that there's some overlap. The number of substances (Figure 6c) and their content (Figure 6d) are different but somewhat redundant. In response to your suggestion, we've removed Figure 6c from the manuscript to streamline the result presentation and enhance clarity. Thanks for helping improve our work (Page 15).
Reviewer 2 Report
Comments and Suggestions for Authors
The manuscript describes the results of metabolomic analyses of herbal tea derived from Artemisia argyi. The plant plays a significant role in Chinese traditional medicine and culture, and it is quite painful to read how it was mistreated in the manuscript.
The methodological problems start with identifying the research material, followed by inadequate measurement methodology and incorrect selection of data processing techniques.
1) Lines 72-73: Whether the plants were obtained commercially or collected from the wild, a qualified botanist should appropriately identify them, and voucher specimens should be deposited.
2) Line 119: The 2ml centrifuge tube does not replace volumetric glass. The samples for LC-MS analyses were not prepared correctly.
3) Line 119: Weighting accuracy should be specified (50 mg ± ???)
4) Lines 137-138: The authors forgot to include the acquisition rate (Hz), the m/z at which the resolution was set, and, most importantly, the automatic gain control (AGC) value, as well as the maximum injection time of the detector.
5) Lines 139-142: The authors should specify how the QC samples were prepared. Were they pooled QCs? Prepared from the entire sample set or group-specific?
6) Lines 124-142: The selected separation gradient does not seem to offer an adequate resolution for advanced metabolomics analyses due to insufficient data points per peak. The authors used 60k resolution for a full scan, which is obtainable on the Orbitrap at an acquisition rate of approx. 5 Hz and probably a maximal acquisition rate of 22 Hz at 15k resolution for MS/MS. The mass spectrometer operates in the scan-fragmentation cycle. Therefore, it would produce 2.5 full scan MS and 55 MS/MS spectra per second. The length of the entire resolving part of the gradient is 420 seconds (between 1 and 8 min runtime = 7 minutes, line 131), corresponding to 1050 data points for the full scan (for both polarities because the authors apparently also used polarity switching) and approx 23k MS/MS data points. Considering the nonsensical value of 8627 metabolites (line 247), we get approx. 2.6 MS/MS spectrum per metabolite, which seems almost reasonable. However, we also have polarity switching, so 1.3 MS/MS spectrum per metabolite per polarity. The reasonable value for the most basic identification would be an average of at least 3-5 MS/MS spectra. In conclusion, the length of the chromatographic analysis should be 3-5 times longer (21-35 min long resolving gradient) to provide reliable enough data for identifying metabolites.
8) Lines 195-197: Although all three libraries used to identify the metabolites are generally suitable, the authors should consider that some contain in-silico predicted spectra. Therefore, the manuscript should be accompanied by a data table containing the basic MS identification data: RT, m/z, adduct type, calculated formula, measurement accuracy in ppm, the primary MS/MS peaks, and, most importantly, the identification confidence level as described in doi: 10.1021/es5002105 or doi: 10.1007/s13361-016-1469-y. Without such a table included in either the main text body or as supplementary material, the manuscript does not meet the minimal requirement for publication.
Furthermore, if the authors discovered that kynurenic acid is one of the crucial biomarkers, it would be customary to include a reference in which its occurrence in the researched plant was confirmed. If there is no such reference, the authors should verify the identity by either co-elution with authentic standards or NMR. Otherwise, the identification is only tentative; the text should indicate that.
7) The author did not describe the data preprocessing (normalization, transformation, scaling) before the multivariate analysis. Preprocessing affects the outcome of these analyses and must be described in detail.
8) Although OPLS-DA is very important for processing metabolomics data, one restriction is that it does not allow for separating more than two classes (doi:10.1016/B978-0-323-85062-9.00009-X). Therefore, it is not suitable for multiclass models. The presented research has four classes of samples...
9) Numerous researchers detected α-thujene in the Artemisia essential oils and volatile samples. However, it is surprisingly missing from the list of volatiles detected by the authors. Such discrepancy should be discussed in the text.
10) The results from HS-GC-IMS and HS-SPME/GC×GC-TOFMS are perfectly amenable to multivariate statistical analyses. Why did the authors not carry out analyses such as PCA on these results?
11) Some figures, particularly 4 and 5, should be prepared in higher resolution or split to increase readability. The font used is too small in size.
Comments on the Quality of English Language
No comments
Author Response
Thank you so much for your letter and the comments from the reviewers about our manuscript (foods-3448036). We tried our best to improve the manuscript and made some changes, which will not influence the content and framework of the paper. Here we did not list the changes but marked them in red in the revised paper. The comments from the reviewers are all valuable and very helpful for revising and improving our paper, as well as the important guiding significance to our research. We have studied comments carefully and have made corrections which we hope to meet with approval. Revised portions are marked in red on the paper. The main corrections in the paper and the responses to the reviewer’s comments are as follows:
Response to reviewer 2
Reviewer #2: The manuscript describes the results of metabolomic analyses of herbal tea derived from Artemisia argyi. The plant plays a significant role in Chinese traditional medicine and culture, and it is quite painful to read how it was mistreated in the manuscript. The methodological problems start with identifying the research material, followed by inadequate measurement methodology and incorrect selection of data processing techniques.
My comments below:
Comment 1. Lines 72-73: Whether the plants were obtained commercially or collected from the wild, a qualified botanist should appropriately identify them, and voucher specimens should be deposited.
Response: We are grateful for your suggestions, we have added the identification information and marked red in the manuscript (Lines 76-80).
Comment 2. Line 119: The 2 ml centrifuge tube does not replace volumetric glass. The samples for LC-MS analyses were not prepared correctly.
Response: We sincerely appreciate the valuable comments. We would like to clarify that 600 µL of methanol containing 2-chloro-L-phenylalanine (4 ppm) was added using a calibrated micropipette to ensure precise volume measurement. The weighed samples (50 mg) and methanol were mixed in a 2 mL centrifuge tube for subsequent processing. To avoid ambiguity, we have revised the methodology to explicitly state the use of a calibrated micropipette and marked red in the manuscript. We appreciate your feedback, which has improved the clarity of our manuscript. (Lines 132-134,136-137)
Comment 3. Line 119: Weighting accuracy should be specified (50 mg ± ???)
Response: Thank you for your comment. The weighing accuracy for the FAA tea samples was 50 ± 1.7 mg, as stated in the revised manuscript (Line 132).
Comment 4. Lines 137-138: The authors forgot to include the acquisition rate (Hz), the m/z at which the resolution was set, and, most importantly, the automatic gain control (AGC) value, as well as the maximum injection time of the detector.
Response: Thanks for your professional review. Regarding the resolution, it was already stated in the manuscript at Line 148 as being set to 60,000 for MS1 and 15,000 for MS2. For the remaining parameters, we have added and marked red in the manuscript (Lines 154-155). We are grateful for your meticulousness, as it has greatly improved the completeness of our manuscript.
Comment 5. Lines 139-142: The authors should specify how the QC samples were prepared. Were they pooled QCs? Prepared from the entire sample set or group-specific?
Response: We thank the reviewer for raising this point. The QC samples were prepared as pooled QCs by combining equal volumes of each extracted sample. This approach ensures representation of the entire sample set, enabling effective monitoring and correction of technical variability during analysis. We have clarified this and marked red in the manuscript (Lines 158-159).
Comment 6. Lines 124-142: The selected separation gradient does not seem to offer an adequate resolution for advanced metabolomics analyses due to insufficient data points per peak. The authors used 60k resolution for a full scan, which is obtainable on the Orbitrap at an acquisition rate of approx. 5 Hz and probably a maximal acquisition rate of 22 Hz at 15k resolution for MS/MS. The mass spectrometer operates in the scan-fragmentation cycle. Therefore, it would produce 2.5 full scan MS and 55 MS/MS spectra per second. The length of the entire resolving part of the gradient is 420 seconds (between 1 and 8 min runtime = 7 minutes, line 131), corresponding to 1050 data points for the full scan (for both polarities because the authors apparently also used polarity switching) and approx 23k MS/MS data points. Considering the nonsensical value of 8627 metabolites (line 247), we get approx. 2.6 MS/MS spectrum per metabolite, which seems almost reasonable. However, we also have polarity switching, so 1.3 MS/MS spectrum per metabolite per polarity. The reasonable value for the most basic identification would be an average of at least 3-5 MS/MS spectra. In conclusion, the length of the chromatographic analysis should be 3-5 times longer (21-35 min long resolving gradient) to provide reliable enough data for identifying metabolites.
Response: Thank you for this valuable comment. We would like to clarify that polarity switching was not used; instead, each sample was analyzed separately in positive and negative ion modes (12 min gradient each), totaling 24 minutes per sample. This approach ensures sufficient data points for reliable metabolite identification. The 12-minute gradient was optimized through internal validation to achieve optimal separation of metabolites in complex samples. We have added the expression and marked red in the manuscript (Lines 141-142). While longer gradients could enhance data quality, our current method, combined with high-resolution Orbitrap Exploris 120 and dynamic exclusion, provides robust results for metabolomics analysis. We will consider the reviewer's suggestion for future studies requiring higher resolution.
Comment 7. Lines 195-197: Although all three libraries used to identify the metabolites are generally suitable, the authors should consider that some contain in-silico predicted spectra. Therefore, the manuscript should be accompanied by a data table containing the basic MS identification data: RT, m/z, adduct type, calculated formula, measurement accuracy in ppm, the primary MS/MS peaks, and, most importantly, the identification confidence level as described in doi: 10.1021/es5002105 or doi: 10.1007/s13361-016-1469-y. Without such a table included in either the main text body or as supplementary material, the manuscript does not meet the minimal requirement for publication.
Furthermore, if the authors discovered that kynurenic acid is one of the crucial biomarkers, it would be customary to include a reference in which its occurrence in the researched plant was confirmed. If there is no such reference, the authors should verify the identity by either co-elution with authentic standards or NMR. Otherwise, the identification is only tentative; the text should indicate that.
Response: We are very grateful to your thoughtfulness. We have added the MS identification data in Table S2 and Figure S2. Regarding the identification of kynurenic acid, we have made the following revisions and additions: We have clearly labeled the identification of kynurenic acid as "tentative" in the manuscript, noting that it is based on LC-MS/MS data and multivariate statistical analysis. Additionally, we have expanded the Discussion section to compare our findings with previous studies, highlighting the distinction between key substances (e.g., flavonoids and volatile oils) and key differential substances (e.g., kynurenic acid). We also explained that these differences may arise from variations in growing conditions, extraction methods, or metabolic regulation. We earnestly appreciate your hard work and hope that the correction will meet with approval. Due to current experimental limitations, we have not yet performed co-elution or NMR validation, but we have included this as a priority for future research. We believe these revisions more accurately reflect our findings and provide a clearer direction for follow-up studies. Thank you again for your insightful feedback. (Lines 344, 350-375)
Comment 8. The author did not describe the data preprocessing (normalization, transformation, scaling) before the multivariate analysis. Preprocessing affects the outcome of these analyses and must be described in detail.
Response: Thanks for your professional review. In the revised manuscript, we have added a detailed description of the data preprocessing steps, including normalization using the LOESS signal correction method based on pooled QC samples to eliminate systematic errors, log2 transformation to reduce the impact of high-intensity peaks and improve data distribution, and autoscaling (mean-centered and divided by the standard deviation) to ensure comparability across variables. These steps were implemented to enhance the reliability and interpretability of the multivariate analysis results. We have added the information in Methods section and marked red in the manuscript (Lines 217-221).
Comment 9. Although OPLS-DA is very important for processing metabolomics data, one restriction is that it does not allow for separating more than two classes (doi:10.1016/B978-0-323-85062-9.00009-X). Therefore, it is not suitable for multiclass models. The presented research has four classes of samples...
Response: We appreciate the reviewer’s insightful comment regarding the application of OPLS-DA in a multiclass setting. We fully acknowledge that OPLS-DA is primarily designed for binary classification, and its direct application to multiclass data has inherent limitations. In our study, OPLS-DA was employed specifically to elucidate pairwise differences among selected key classes, thereby identifying potential biomarkers that are critical to our research objectives. To address the limitations associated with the multiclass context, we complemented the OPLS-DA analysis with additional multivariate statistical methods, including PCA and PLS-DA. These approaches provided a more comprehensive overview of the overall data structure and classification performance across all four sample classes. The integration of these methods ensures that our findings are robust and that the conclusions drawn are supported by multiple lines of evidence. In the revised manuscript, we will include a more detailed discussion of the analytical strategies employed, along with explicit descriptions of the results obtained from each method (Lines 271-277, 286-295, 303-309, 321-325, 514-517). We believe that this clarification will help readers better understand our approach and the rationale behind the use of OPLS-DA in conjunction with other complementary analyses. Thank you again for your valuable feedback, which has significantly contributed to the improvement of our manuscript.
Comment 10. Numerous researchers detected α-thujene in the Artemisia essential oils and volatile samples. However, it is surprisingly missing from the list of volatiles detected by the authors. Such discrepancy should be discussed in the text.
Response: Thank you for pointing out this discrepancy. We would like to clarify that α-thujene was not detected in the HS-GC-IMS analysis, but it was identified as a significant differential metabolite in the HS-SPME/GC×GC-TOFMS analysis (as shown in line 21 of Figure 7c). This difference may be attributed to the varying sensitivity and selectivity of the two analytical techniques. To address your concern, we have added a discussion of α-thujene in the revised manuscript (Lines 526-535), highlighting its potential significance in the context of our findings.
Comment 11. The results from HS-GC-IMS and HS-SPME/GC×GC-TOFMS are perfectly amenable to multivariate statistical analyses. Why did the authors not carry out analyses such as PCA on these results?
Response: Thank you very much for double-checking and pointing this out. PCA analysis on the HS-SPME/GC×GC-TOFMS data was already conducted and described in Lines 498-506 of the original manuscript. As per the reviewer's suggestion, we have now added PCA analysis for the HS-GC-IMS data, marked in red in the revised manuscript (Lines 423-427, Figure 6a). These additional analyses enhance the robustness of our findings. We value the reviewer's input, which has greatly improved the study's completeness.
Comment 12. Some figures, particularly 4 and 5, should be prepared in higher resolution or split to increase readability. The font used is too small in size.
Response: We sincerely thank the reviewer for pointing out this issue. We have carefully recreated Figures 4 and 5 in higher resolution, and split Figure 4 to ensure that all details are clearly visible. Additionally, we have increased the font size in the figures to improve readability. We hope these changes address the concerns raised, and we appreciate the reviewer's valuable feedback, which has helped enhance the quality of our manuscript. (Pages12-14).
Reviewer 3 Report
Comments and Suggestions for Authors
The manuscript entitled "A Comprehensive Metabolomic Analysis of Volatile and Non-Volatile Compounds in Folium Artemisia argyi Tea from Different Harvest Times" presents a metabolomic study of the volatile and non-volatile compounds in Folium Artemisia argyi tea grown at different harvest times. Thousands of compounds were identified, several of which are crucial for understanding the biosynthetic profile and the antioxidant capacity of plant compounds at specific times of the year. The manuscript is well-written, and the study was conducted rigorously, making it scientifically relevant.
I suggest minor corrections
- Describe the geographic coordinates where the plants were collected
- Add the chemical structures of some compounds and chromatograms in the manuscript or in the supplementary material.
Author Response
Thank you so much for your letter and the comments from the reviewers about our manuscript (foods-3448036). We tried our best to improve the manuscript and made some changes, which will not influence the content and framework of the paper. Here we did not list the changes but marked them in red in the revised paper. The comments from the reviewers are all valuable and very helpful for revising and improving our paper, as well as the important guiding significance to our research. We have studied comments carefully and have made corrections which we hope to meet with approval. Revised portions are marked in red on the paper. The main corrections in the paper and the responses to the reviewer’s comments are as follows:
Comment 1. Describe the geographic coordinates where the plants were collected
Response: We are grateful for your suggestions, we have added the geographic coordinates information and marked red in the manuscript (Lines 73-76).
Comment 2. Add the chemical structures of some compounds and chromatograms in the manuscript or in the supplementary material.
Response: Thanks for your professional review. We have added the chemical structures of some compounds and chromatograms in Figure S3 and Figure S4 respectively.